# Optimized criteria for locomotion-based healthspan evaluation in *C. elegans* using the WorMotel system

**Areta Jushaj**[1], **Matthew Churgin**[2], **Bowen Yao**[2], **Miguel De La Torre**[2], **Christopher Fang-Yen**[2], **Liesbet Temmerman**[1] *

**1** Animal Physiology and Neurobiology, Department of Biology, KU Leuven, Leuven, Belgium, **2** Department of Bioengineering, University of Pennsylvania, Philadelphia, United States of America

* Liesbet.Temmerman@kuleuven.be

## Abstract

Getting a grip on how we may age healthily is a central interest of biogerontological research. To this end, a number of academic teams developed platforms for life- and health-span assessment in *Caenorhabditis elegans*. These are very appealing for medium- to high throughput screens, but a broader implementation is lacking due to many systems relying on custom scripts for data analysis that others struggle to adopt. Hence, user-friendly recommendations would help to translate raw data into interpretable results. The aim of this communication is to streamline the analysis of data obtained by the WorMotel, an economically and practically appealing screening platform, in order to facilitate the use of this system by interested researchers. We here detail recommendations for the stepwise conversion of raw image data into activity values and explain criteria for assessment of health in *C. elegans* based on locomotion. Our analysis protocol can easily be adopted by researchers, and all needed scripts and a tutorial are available in S1 and S2 Files.

**Data Availability Statement:** All relevant data are within the paper and its Supporting Information files. The series of images accompanying the tutorial for data analysis have been made available

## Introduction

While a bliss for individuals in good health, the continued increase in human life expectancy is also associated with an increased prevalence of age-related diseases, warning our societies to tackle this socio-economic challenge. Therefore, amelioration of the quality of life in aged populations will be an important task in years to come.

It is well understood in the research field that the concept of being healthy is much more ambiguous than the concept of being alive, and different individuals have a different perception of what is understood as 'being healthy' [1]. Healthspan is generally described as the period in life during which the organism is in good health and free from disease [2]. It is immediately clear that this fluid definition reflects a similar lack of consensus amongst researchers, which translates to a variety of proposed parameters for healthspan evaluation. In human clinical settings, grip strength, gait analysis and ability to perform daily tasks (*e.g.* bathing) are often used as criteria for good health [3–5].

at https://www.ebi.ac.uk/biostudies/studies/S-BSST313#.

**Funding:** All authors received project funding from the European Union's Horizon 2020 research and innovation programme (633589). This work was supported by the KU Leuven Research Council (C14/15/049) to LT. AJ received travel support from the FWO Flanders (V426816N and K218918N) and Junior Mobility Program at KU Leuven (JUMO/16/021 and JUMO/18/009). The funders had no role in study design, data collection and analysis, decision to publish, or preparation of the manuscript. Strains used in this study were provided by the Caenorhabditis Genetics Center (CGC), which is funded by National Institutes of Health (Office of Research Infrastructure Programs Grant P40 OD010440). European Union's Horizon 2020 research and innovation programme: https://ec.europa.eu/programmes/horizon2020/en). KULeuven Research Council: https://admin.kuleuven.be/raden/en/research-council FWO: https://www.fwo.be/. Junior Mobility Program at KU Leuven: https://www.kuleuven.be/personeel/careercenter/youreca-career-center/yourecaENG/youreca-internationalmobility.

**Competing interests:** The authors have declared that no competing interests exist.

To find interventions that affect aging, considerations in cost and time-efficiency have led to the use of different model organisms. The nematode *C. elegans* is a well-established model for aging with the advantages of a short lifespan and ease of cultivation. Work in this model indicates that a longer lifespan does not always correlate with proportional increases in healthy life [6–9], reaffirming the notion that understanding how organisms can age healthily is important.

As summarized by [6], several physiological and functional parameters that change with age can be studied in *C. elegans*, such as lipofuscin accumulation or pharyngeal pumping. Among these, the most powerful predictor of longevity seems to be movement [8,10–13]. Similar to humans, the ability of *C. elegans* to move diminishes with aging [14], as they decline towards a state of frailty where they are only able to move their head, characteristic of late phases of life. Research into *C. elegans* aging is often challenging due to labor-intensive follow-up of experiments and the collection of longitudinal data at the population level, rather than at the level of the individual. To address these impediments, several groups developed semi-automated systems that bypass (some of) these issues and assess movement longitudinally in aging animals [7,15–18]. All these systems rely on longitudinal imaging of either individual [7,12,13,16,19] or populations of [17,18,20,21] worms, after which image processing and analysis are used for determination of lifespan and activity decline. While relying on similar principles, these systems differ in high-throughput potential, detail in image acquisition, the way the worms are stimulated to move and whether populations or individuals are studied.

The WorMotel [22] allows longitudinal measurements of activity in 240 individual animals simultaneously. Time-lapse images of the aging worms are used to quantify their movement for life- and healthspan determination. Because 240-well plates are typically imaged for only 20 minutes per day, one imaging station can collect data for thousands of individuals each day. Due to the aspired throughput of this system, having a clear-cut analysis protocol that can distinguish phenotypes, is crucial. In terms of lifespan, this is straightforward, but because of the ambiguous definition of healthspan, criteria to determine whether an animal is healthy or not are currently lacking. While basic programming tools to calculate movement based on images taken with the WorMotel system were developed and reported [7], users are faced with several choices for image processing parameters during data analysis. These include deciding on a time interval for activity calculation, condensing data of the intermittent monitoring periods into data points along longitudinal activity curves, setting a reasonable threshold for health, and considering the applicability of chosen settings to long- and short-lived populations. A systematic study on how these choices affect outcomes such as individual life- and healthspan has not been carried out. We therefore aimed to determine the most robust choices for straightforward selection of interventions that may affect healthy ageing.

We performed a proof of concept study on data of wild-type, long-lived *daf-2* RNAi-treated and short-lived *daf-16* RNAi-treated animals. This work provides insight into recommended standard settings and can serve as a basis for users of the WorMotel to tune their own data processing choices and highlight specific behaviors of interest.

## Materials and methods

### Strains, maintenance and worm synchronization

In this study, wild-type N2 animals fed on *E. coli* OP50 were used. Strain maintenance and experiments were performed at 20˚C. Mixed cultures were bleached and eggs were collected by standard procedures [23].

## WorMotel plate preparation and RNAi

WorMotel plates were prepared as described by [7]. We used *E.coli* strain HT115 transformed with the L4400 vector containing no (= control), *daf-2* or *daf-16* RNAi constructs as derived from the Ahringer library [24]. Bacterial strains were grown at 37˚C for a minimum of 12 hours, whereafter 1 mM IPTG was used for induction (2 hours, 37˚C). To minimize effects of diet, equal amounts of bacterial solutions were seeded onto WorMotel wells, and potential positional effects were minimized by doing so according to a quadrant design (60 wells per strain–one control and three test strains per plate). Hatched N2 L1 stage worms were grown on Nematode Growth Medium (NGM) plates containing carbenicillin (50 μg/ml) seeded with *E. coli* HT115 containing the (empty) L4440 vector and reared at 20˚C for 48 hours. At late L4 stage, worms were sorted using a COPAS Biosort (Union Biometrica) onto WorMotel plates.

## Image processing and parameter extraction

Each plate was monitored daily for 20 minutes with an Imaging Source DMK 23GP031 camera (2592 x 1944 pixels) equipped with a Fujinon lens (HF12.5SA-1, 1:1.4/12.5 mm, Fujifilm Corp., Japan) as previously described [7]. We used IC Capture (Imaging Source) to acquire time-lapse images through a gigabit Ethernet connection, this over a period of approximately 40 days. Images were taken every five seconds and a five-second blue light stimulation was applied at minute 10. For the blue light stimulation, three high-power LEDs (at a current of 20 A, Luminus PT-121, Sunnyvale, CA, irradiance at plate 1.2 mW/mm$^2$) were used. Image subtraction with a custom-made MATLAB script was performed, where for every captured image (~ 120 pictures post blue light stimulation) pixel value intensity changes are calculated in comparison with an image preceding it by a defined interval (not necessarily the preceding image in the series). For this study, pixel differences were calculated for intervals of 5, 20, 60, 80, 100, 150, 200, 250, 300 and 540 seconds (S1 File). Calculations were executed according to [7]. Briefly, for each set of two compared images, a difference image was calculated and divided by the average pixel intensity between the two images to generate normalized maps of pixel value intensity change. Incorporation of noise was reduced by consecutively applying (i) a Gaussian smoothing filter (standard deviation of one pixel) and (ii) a binary threshold of 0.25 to the difference image [7]. The total number of pixel locations changed on the resulting image was then used as a measurement for activity. We always worked with post-stimulation pixel difference data (collected minutes 10–20), since it has been shown that spontaneous activity is a confounded readout [7]. Moreover, we observed that stimulated activity leads to a more reliable assessment of lifespan in aged animals, as they tend to show less spontaneous movement (Fig 1).

To convert the imaging data to a single value per condition per day, we considered several options that represent different ways of looking at the animal's ability to move (Fig 2). For this, for all daily pixel difference data series the median, 99$^{th}$ percentile (or 'maximal activity'), average of all values within the range defined by [95$^{th}$ to 99$^{th}$] percentiles (further referred to as 'peak activity') and integral (corresponding to area under the curve of Fig 2A) were calculated (Fig 2A).

## Overall variation

Overall variation was calculated as:

$$overall\ variation = \frac{\sum_{i=1}^{N}\left(\frac{|a_{i+1}-a_i|}{<a_{1\to N}>}\right)}{N}$$

with N the total number of days the worm was monitored as alive, $a_i$ the activity value on day i

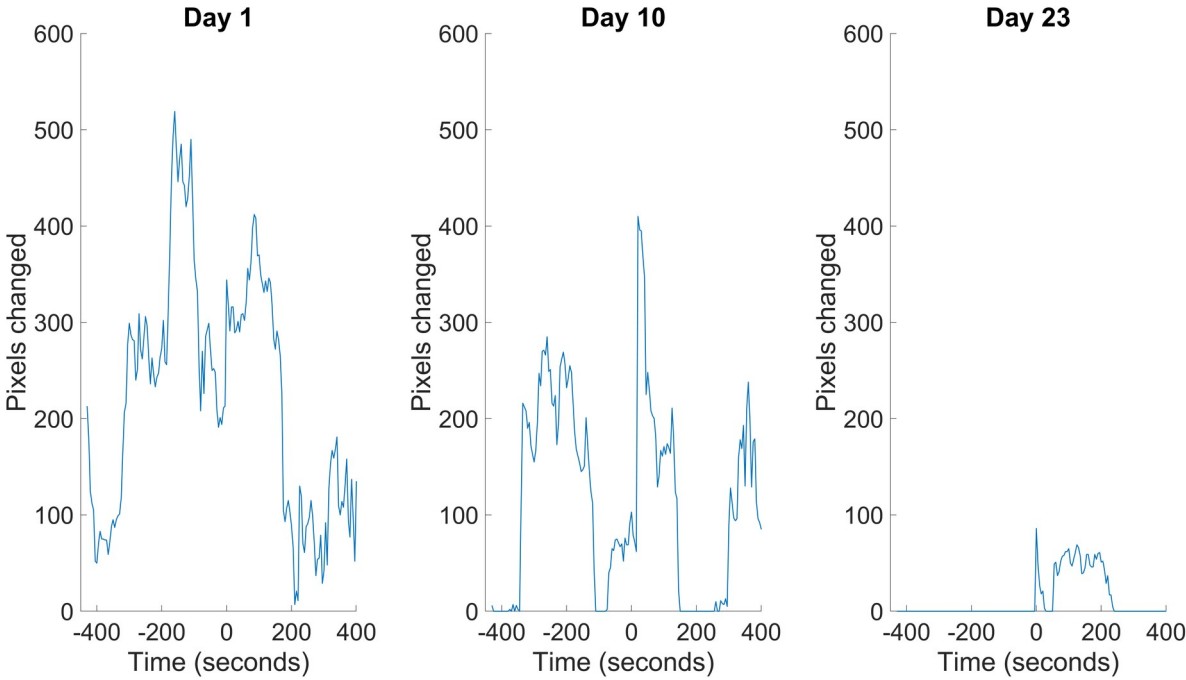

**Fig 1. Data collected after blue light stimulation are most suitable for quantification of lifespan.** Data are from one representative animal. Blue light stimulation (at "time 0" of the imaging interval) is crucial to ensure accurate lifespan determination, especially in older animals, which typically display little or no spontaneous movement within the 20 min imaging interval.

for that worm, and $<a_{1\to N}>$ the average activity of the animal over its lifetime. As such, an individual's overall variation reflects its average change in activity per day. In short, for each worm, all differences in activity between each two consecutive days are summed. This value is normalized by dividing it by the average activity of the worm over its lifetime, a necessary step for comparison of metrics with different magnitudes (*e.g.* median *vs* integral). For ease of comparison, a daily value is obtained by dividing by the number of observations (note that this is not essential for interpretation). For each worm, overall activity values were calculated based on median, maximal activity, peak activity and integral (Fig 2) input values.

## Variance of movement Z-score

Each individual's movement *Z*-score as a function of time is defined as:

$$Z - score = \frac{a_i - \mu_{a_i}}{\sigma_{a_i}}$$

where $a_i$ is the activity value of the worm on day i, $\mu_{a_i}$ is average activity of the population on day i and $\sigma_{a_i}$ is the standard deviation of the population on that day. A *Z*-score therefore reflects how different an individual worm is from the population, and this for each day of its life. When calculating the variance of this *Z*-score for each individual worm, a value is obtained that reflects the magnitude to which the longitudinal activity profile of an individual worm deviates from that of the (simultaneously alive part of the) population.

## Lifespan and health determination

The lifespan of each worm was always determined as the last day when the worm showed a daily peak activity above 5 pixels changed, as described in [7].

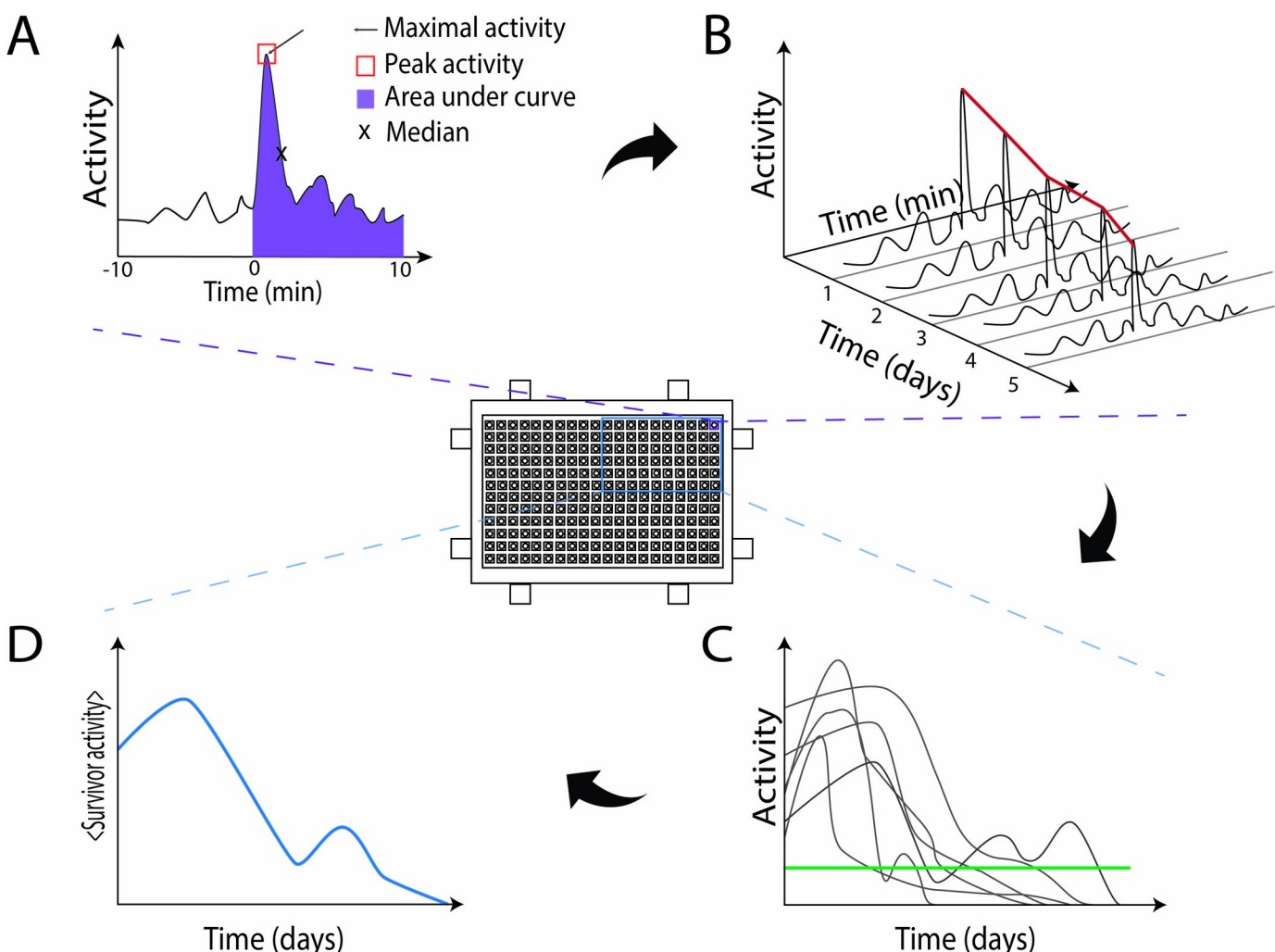

**Fig 2. Overview of data analysis.** The activity of a single worm during one monitoring time (one day) can be summarized in different ways, relying on **(A)** the median, 99th percentile (also: 'maximal activity'), average of all values between the 95th to 99th percentiles (also: 'peak activity', red box) or integral (purple shading) values of the pixel difference data. **(B)** This process can be repeated for different days for the same worm, unveiling how activity (here: peak activity) changes over a lifetime. **(C)** This analysis is performed for all the worms of the population, based on which **(D)** the average survivor activity of worms belonging to the same population can be calculated. Worms showing an activity above the green threshold (panel C) are considered healthy—see main text for details.

To determine whether an animal was healthy or not, we correlated blinded manual assessments of health with calculated pixel differences for a representative set of animals. Locomotive health was empirically evaluated by three independent scientists for blinded activity movies of 24 randomly chosen worms, aimed to represent two worms per genotype per plate (randomly selected from each population). These 24 animals to be evaluated for each monitoring time lead to a total of 715 manual assessments, all executed in triplicate. Each scientist could assign quality of movement upon blue light stimulation to one of five categories: (1) very fast: the animal moved multiple (>2) body lengths, (2) fast: the animal moved 1 to 2 body lengths, (3) medium fast: the animal continuously moved, but within body length, (4) slow: the animal did move within body length, but was then inactive, and (5) inactive: the animal did not move. One worm had to be excluded from the analysis because it did not belong to a genotype discussed in this study, leading to a total of 23 studied worms.

For our analyses, we distinguish 'total days of health' from healthspan. Total days of health (TDH) refer to the total number of days—not necessarily consecutive—for which an individual displays an activity greater than the 160 pixel difference threshold (see Results). Healthspan (HS) is defined as the very last day when an individual's activity is above said threshold. Obviously, the value for 'total days of health' is always lower than that of healthspan. We further define the health ratio (HR) of individual worms as the 'total days of health' divided by the total days of lifespan. Alternatively, healthspan ratio (HSR) can be calculated as the ratio of healthspan *vs* lifespan.

As a final metric for health interpretation, we calculated the definite integral of the average activity of the population by approximation through the trapezoid rule. This value approximates the area under the curve of the average activity of the studied population, whose shape depends on the genotype [7,13].

### Statistical analysis and graphical representations

Graphical representations and statistical tests–regarding normality (Shapiro-Wilk), significance of population differences (Kruskal-Wallis or ANOVA) and correction for multiple testing (Tukey-Kramer)–were run using MATLAB®. Linear correlation between different metrics for the same worm, at the same time point, was assessed based on least-squares fit and calculated using MATLAB®.

## Results

In recent years, diverse research teams worked towards alleviating the labor-intensive aspects of *C. elegans*-based studies of longevity and aging. Amongst the developed semi-automated platforms [17,18,20], the WorMotel [7] stands out for its capacity to collect data on thousands of individuals on a daily basis, adding to its appeal as a medium- to high-throughput screening solution for studies of aging. To facilitate the adoption of this system for fast evaluation of high numbers of interventions, we here evaluate the data analysis workflow and discuss analytical decisions made during the process of life- and healthspan analysis.

We collected data of wild-type animals reared under control conditions (marked 'empty vector' (EV)) or treated with *daf-2* or *daf-16* RNAi. The exact effects of genetic interventions on the lifespan of *C. elegans* vary somewhat in high-throughput [17] screens and between different labs [25], but *daf-2* consistently leads to longevity, while *daf-16* consistently shortens lifespan. To reflect the expected variation when different labs use the WorMotel platform, we used data of four completely independent experiments that represent a large plate-to-plate variability, run over a period of 4 months. For one of the four independently executed experiments (Fig 3), the study period was terminated before it could capture all deaths of *daf-2* RNAi-treated animals, as can happen for long-lived interventions when evaluated in high-throughput screens.

Data were obtained and processed as described in 'Materials and Methods'. During this procedure, the analyst faces several choices that may influence the final results. We evaluated these potential choices at each step, to propose a workflow minimizing variation while remaining applicable to and control, and short-, and long-lived phenotypes.

### Daily peak activity is the more robust activity parameter

For calculations of life- and healthspan, any worm's daily activity trace needs to be converted to a single value per worm per day (Fig 2). There are different ways to do this, representing slightly different biological perspectives.

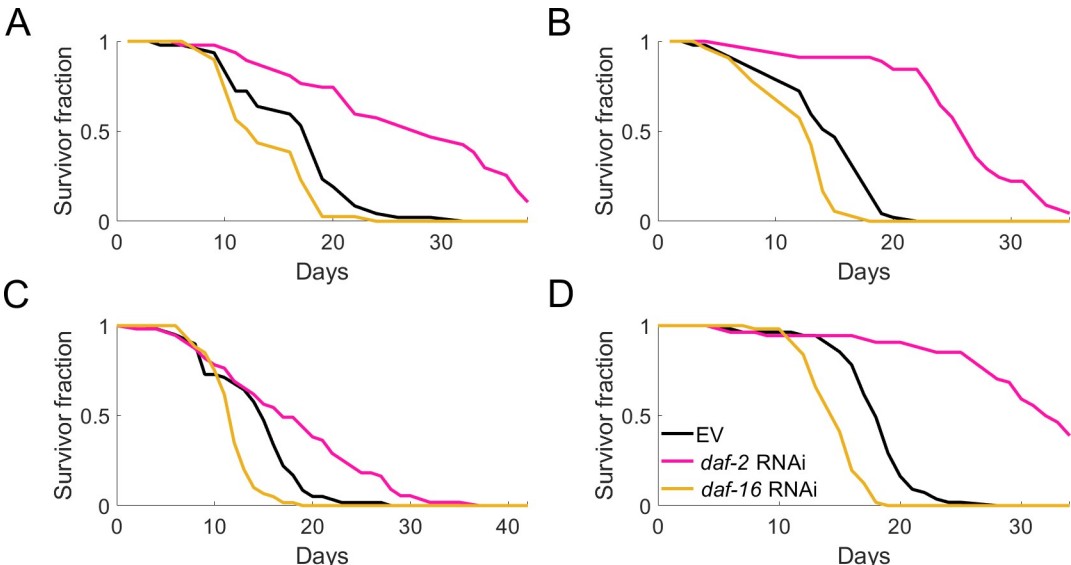

**Fig 3. Independent experiments comparing control, *daf-2* and *daf-16* RNAi-treated populations show inter-experiment differences yet adhere to expected relative survival changes.** To capture expected variation between possible end-users, four entirely independent experiments were performed, where many factors—including the robotic setup—could have contributed to differences in absolute effect size. Experiments I-IV shown in panels A-D.

One way to define the worm's activity, is by taking the median of the worm's response activity values (Figs 2A and 4A). This value is less sensitive to fluctuations due to noise than the more commonly used mathematical average, therefore it is an expectedly more robust way to define the worm's "average" response over the studied time interval. Alternatively, one can look at the peak response, representative of the animal's maximal response to the stimulus. To

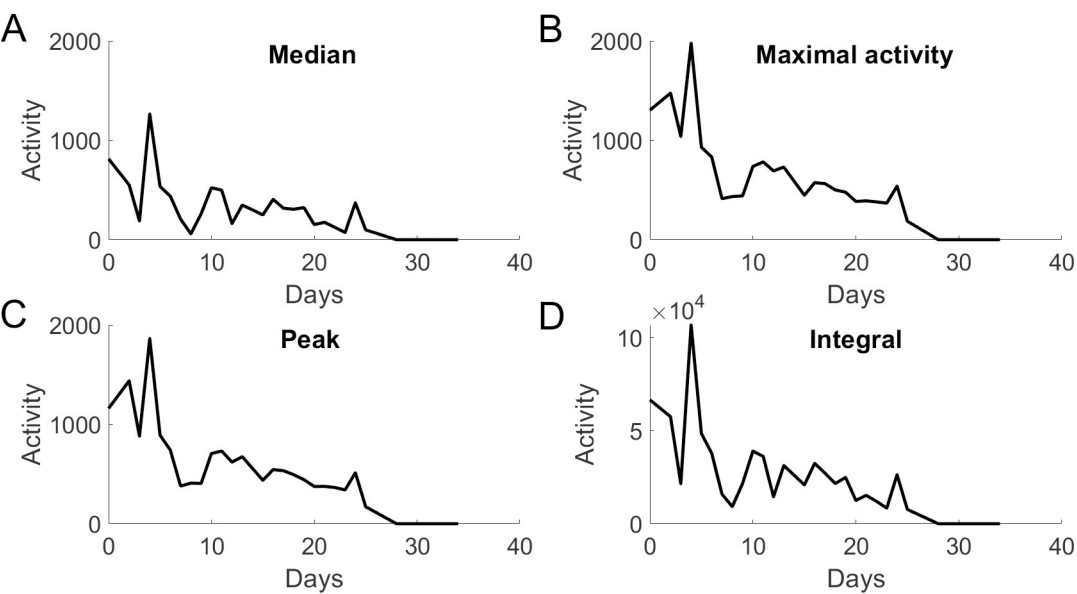

**Fig 4. All metrics used to produce an activity trace reflect the inherent day-to-day variation.** Traces of a representative *daf-2* RNAi-treated worm, constructed based on **(A)** the median, **(B)** maximal activity, **(C)** peak activity or **(D)** integral—see main text.

extract this information, we relied on 99[th] percentile (maximal activity) (Figs 2A and 4B), but also on the average of all values within the range defined by the [95[th] to 99[th]] percentiles (peak activity) (Figs 2A and 4C), as this latter value again should be slightly less prone to outliers or noise than the 99[th] percentile. Finally, the integral, *i.e.* the area under the curve, is indicative of the worm's overall capability to move and maintain that activity (Figs 2A and 4D). For example, one could expect an older worm to still quickly respond to the stimulus, but also return fast to very low activity levels, whereas a younger worm might, upon stimulation, keep up the elevated movement for a longer time. Both median and integral values would capture such a difference better than maximal or peak values would.

Possible interdependency of these parameters can easily be assessed by a simple correlation analysis. Linear regression analysis suggests that peak and maximal values correlate strongly on one hand, and median values clearly correlate with the integral values on the other hand (S1 and S2 Figs, and S1 Table). All other correlations are much weaker, suggesting that these four parameters reflect two interpretations of the daily activity profiles: peak/maximal *vs* median/integral (S1 and S2 Figs, and S1 Table).

The worm's activity trace over its lifetime is ultimately used to determine life- and healthspan of the animal. Due to sparse sampling in high-throughput settings (such as monitoring once or twice per day for a short period of time), however, activity traces are discontinuous and their fluctuation results from a combination of biological and technical influences [7]. For each worm, building longitudinal activity curves based on each of the studied parameters–*i.e.* median, peak, maximal or integral activity (example shown in Fig 4)–will therefore unveil small differences in day-to-day variation that reflect differences between parameters in capturing biological and/or technical sources of variation. The ideal parameter minimizes technical viariation while correctly reflecting biological variation.

We tested the effect of parameter choice based on two assumptions: (i) aging is accompanied by a gradual activity decline on the slow timescale [14], and (ii) variation of the population average may reflect true biological variation. For this, we sought to minimize (i) 'overall variation' and (ii) variance of movement *Z*-score, as defined in Methods. Briefly, for each worm, the overall variation measures its average day-to-day variation over its lifetime, while the variance of its *Z*-score represents how different this individual's activity was from that of the entire population. Using the peak or maximal activity to define the worm's daily activity showed significantly lower overall variation (Fig 5 and S2 Table). We found this to be consistent over genotypes and time intervals, except for the 540 s interval, where results are similar for all tests (S3 Fig). The variance of the *Z*-score never differed for any of the tested parameters at any time interval (Fig 6 and S3 Table). Together, this indicates that although the choice of parameter does not significantly affect population spread (*i.e.* worms do not differently deviate from populations depending on the chosen parameter), the peak or maximal values do create smoother activity traces (lower overall variation) compared to those generated from median or integral values.

Based on these considerations, we opted for the peak activity as the activity value of choice, preferring it over the maximal activity based on its ability to better buffer possible outliers and therefore, it being an expectedly more accurate representation of true biology when compared to the maximal value. The peak activity is used for the remainder of this study.

## Longer time intervals are better suited to determine lifespan

It could be possible that the time interval used for daily pixel difference calculations (*i.e.* image subtractions) affects ultimate decisions on lifespan. To test this, we first averaged daily activities of surviving worms of the same genotype for each day of the population's lifespan (Fig 7

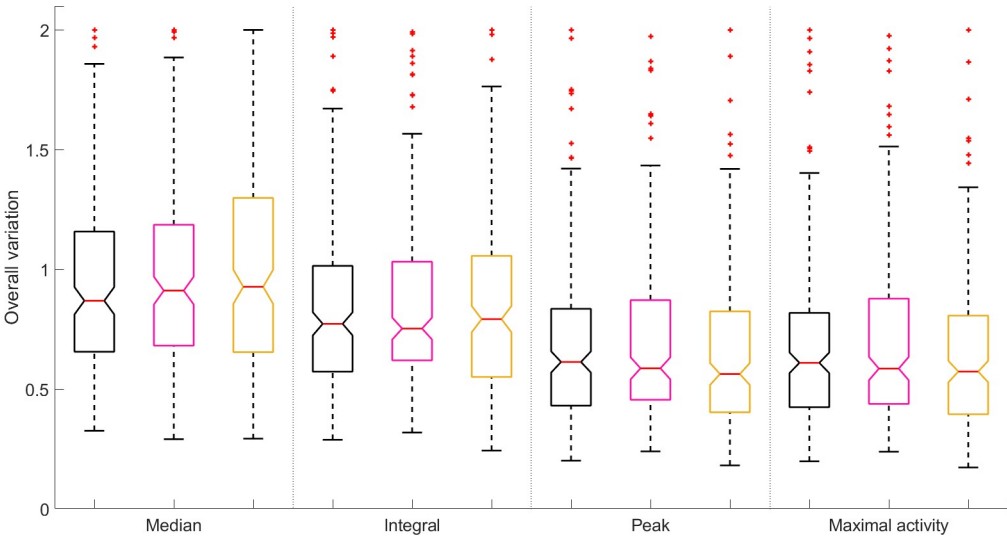

**Fig 5. Data based on peak or maximal values lead to lower overall variation than those based on median or integral values.** Data for activity values based on daily pixel differences of 100 second intervals (other intervals: S3 Fig) for control (EV, black), *daf-2* RNAi (pink) and *daf-16* RNAi (yellow) show similar trends. Box plots based on individual worm data from all worms of the same genotype across all experiments. Analysis on an individual experiment basis leads to the same conclusion (S4 Fig).

for experiment I, S7 Fig for experiments II-IV). Higher intervals lead to higher activity values, which may be especially important for animals with *daf-2*-like longevity, showing lowered but consistent movement during the later phases of life [13]. As can be expected, longer time intervals are able to saturate pixel difference values such as those recorded in the early phases of life, with the highest average activities for most populations peaking around 400 activity units. It is

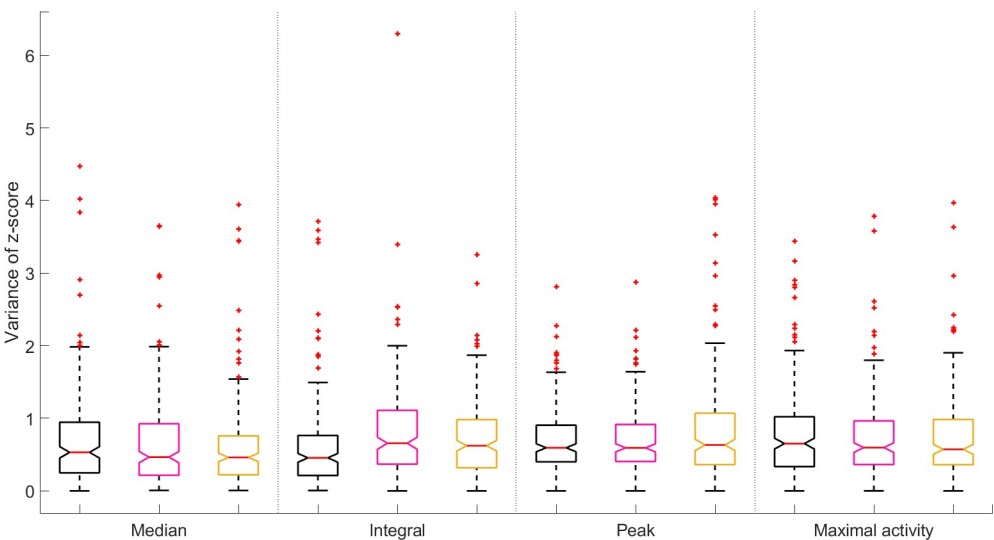

**Fig 6. Choice of activity parameters does not affect variance of Z-score.** Data for activity values based on daily pixel differences of 100 second intervals (other intervals: S5 Fig) for control (EV, black), *daf 2* RNAi (pink) and *daf-16* RNAi (yellow) show similar trends. Box plots based on individual worm data from all worms of the same genotype across all experiments. Analysis on an individual experiment basis leads to the same conclusion (S6 Fig).

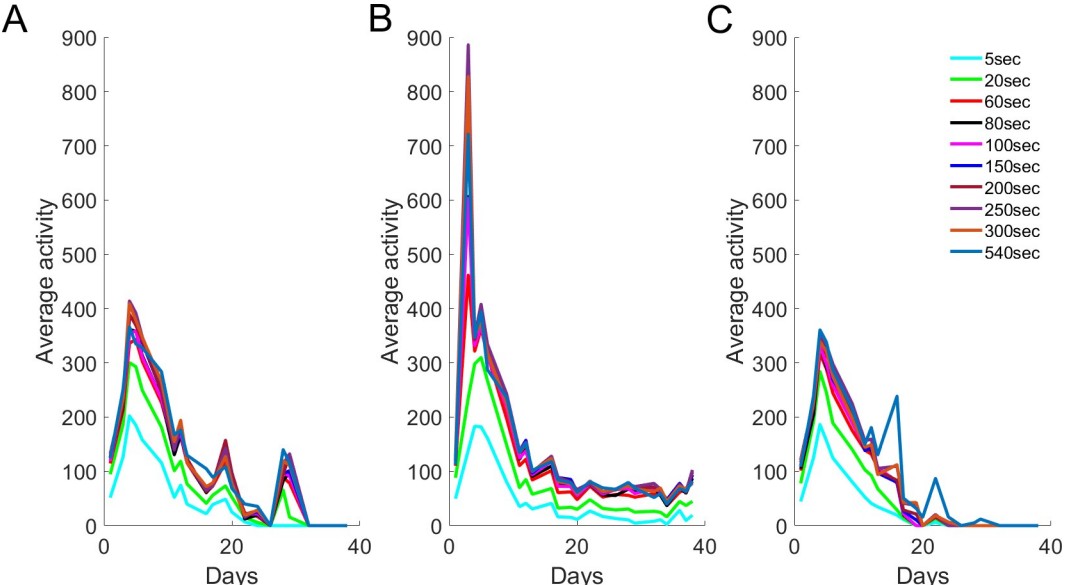

**Fig 7. Average survivor activity is always higher for longer time intervals. (A)** Control populations display an activity decline in line with [7]. **(B)** The typical 'twilight tail' [13] or 'gerospan' [26] is observed in *daf-2* RNAi-treated populations, where animals maintain low-level activity for the majority of their extended life. **(C)** In contrast, the activity of *daf-16* RNAi-treated populations decreases slightly faster than that of controls.

important to keep in mind that fewer worms are alive at later time points, therefore the recorded survivor activity relies on fewer data points as time progresses (Fig 7).

The choice of time interval does not influence the determination of lifespan for control or short-lived (*daf-16* RNAi-treated) populations (Fig 8 and S4 Table). However, in the case of

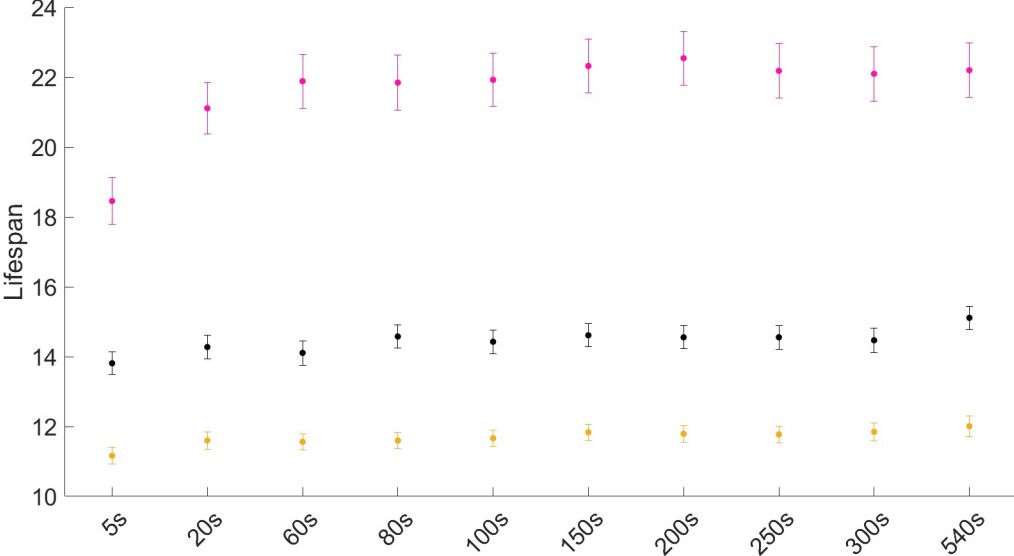

**Fig 8. Lifespan of the long-lived condition is most sensitive to the choice of time interval.** Population averages as calculated for control (black), *daf-2* (pink) and *daf-16* (yellow) RNAi-treated populations (error bars: standard error of the mean), when different time intervals are used for determination of lifespan. Time intervals of $\geq 60$ seconds are advisable.

*daf-2* RNAi-treated animals, time intervals do affect the lifespan decision (S4 Table). Here, the calculated average lifespan reaches a plateau for intervals as of 60 seconds (Fig 8). Similar trends are true for individual experiments (S8 Fig). Overall, time intervals ≥ 60 seconds are acceptable for robust lifespan determination.

## Defining individual health

The ultimate goal of this analysis is to facilitate the search for conditions that affect life- and healthspan via medium- to high-throughput screens. Whereas lifespan is based on a binary measurement (the worm is either alive or not), healthspan is a nebulous concept. To identify interventions that affect health in large screens, however, a simple 'health threshold' that allows a similar binary decision, would nevertheless be helpful.

To test whether such a threshold can be found, we determined which observed pixel difference values correspond to which qualitative assessments of health. For this, we defined five categories describing an animal's movement (very fast—fast—medium fast—slow—inactive, see Methods). Blinded evaluations of 694 activity movies collected from 23 animals over their entire lifespans then allowed to assign each movie to one of these categories. We decided that animals in the slow or inactive categories—*i.e.* barely moving, or not at all (see Methods)—are unhealthy. When linking the WorMotel-calculated pixel difference to the categorical value of each data point (Fig 9A), this analysis showed that despite some overlap between qualitative categories, decreasing pixel differences correspond to decreasing locomotive health. Therefore, the most suitable threshold value should maximize the number of truly healthy worms in healthy (very fast—fast—medium fast) categories, while maximizing the number of truly unhealthy worms in the unhealthy categories (slow–inactive, excluding data for dead animals). A first analysis including all data, showed that such a threshold can be found around 177 pixels changed (S9 Fig). Further refining to best differentiate the 'medium fast' and 'slow' animals, reveals a threshold to be set at approximately 160 pixels different (Fig 9B), an observation that

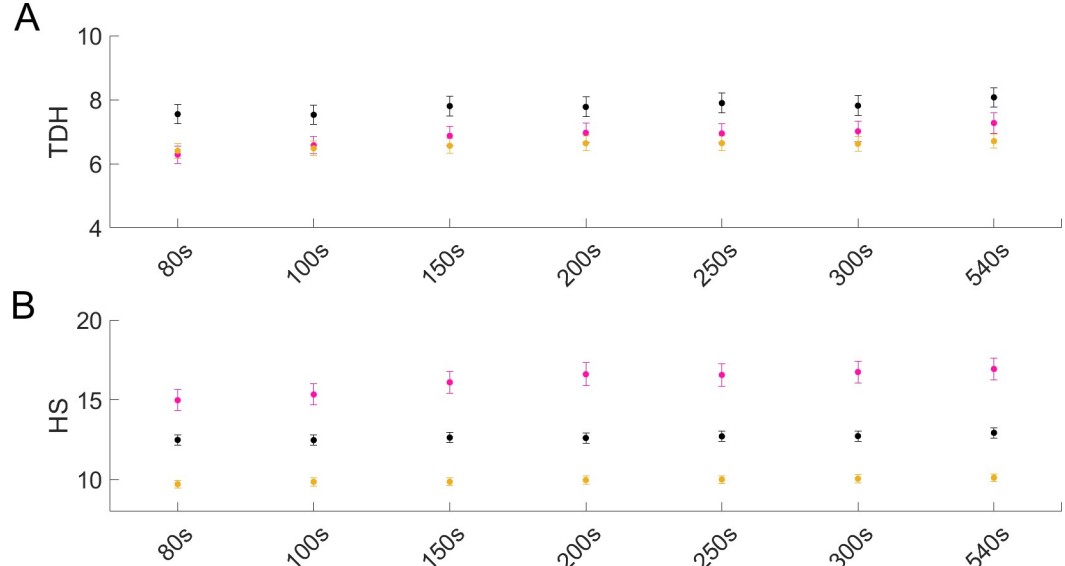

**Fig 9. Decreasing pixel differences caused by worm movement correspond to decreasing locomotive health. (A)** Each dot in the figure represents the pixel difference (calculated by WorMotel analysis) for a single observation that was assigned to one of five qualitative categories (x-axis). **(B)** Above a pixel difference of 160, most animals are scored as healthy (~medium fast movement) by operators, whereas the majority of animals below this threshold are considered less healthy (~slow movement). Black line: fraction of animals in the 'slow' category with a pixel difference value < x-axis value; red line: fraction of animals in the 'medium fast' category with a pixel difference value >x-axis value.

holds true when analyzing individual plates (S10 and S11 Figs). We defined a worm as healthy when it showed an activity value above 160 pixels of difference, which is on the lenient side of the pixel difference options for threshold-based, binary (yes-no) assessment of health.

Building on this threshold, we define 'total days of health' (TDH) as the total number of days on which the animal showed an activity higher than 160 pixels, and the healthspan as the last day of its life for which this was true. One can then calculate health(span) ratios as TDH/lifespan or healthspan/lifespan, to reflect the proportion of its life an animal can be considered healthy.

### Longer time intervals also suit the assessment of health status

As is true for lifespan, the choice of time interval between analyzed images (Fig 8, S4 Table and Methods) affects the ability to quantify an animal's activity, hence, health. We tested the effect of time-interval on the quantification of TDH and HS, based on the threshold for health being 160 pixels changed. As time intervals below 80 seconds are unadvisable for lifespan calculations, we opted to look at the time intervals ranging from 80 to 540 seconds included in this study. Within this range, the choice of time interval does not affect the quantification of TDH and HS for any of the genotypes (Fig 10 and S4 Table). For health(span) ratios, the effects of time interval are also less outspoken as these will influence TDH&HS *vs* LS values in similar ways.

Taken together (Figs 8 and 10), our data show that time intervals of >60 seconds are suitable to analyze WorMotel data of diverse conditions. We propose 100 seconds as the ideal compromise between improved activity detection and number of data points collected during one monitoring period, as the latter decreases for increasing time intervals.

### TDH, HS and integrated activity of the population together aid in interpreting health

Whereas the concept of healthspan (HS) fits the hypothesis of gradual activity decline over aging very well, it is susceptible to severe misinterpretation of longitudinal health in a number

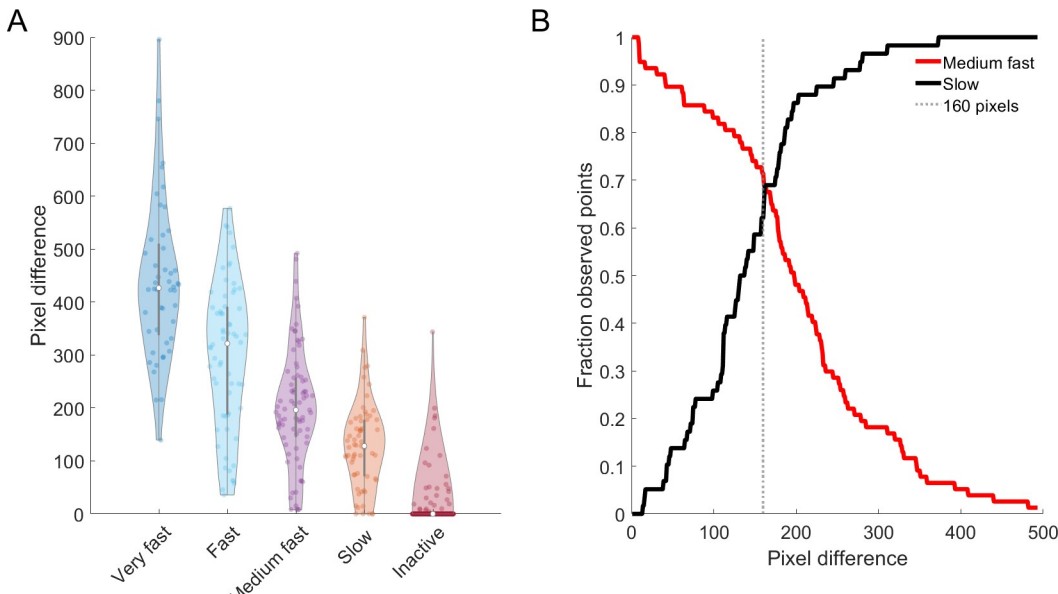

**Fig 10. Total days of health and healthspan quantifications are not sensitive to the choice of time interval within the 80–540 seconds range.** Population averages for control (black), *daf-2* (pink) and *daf 16* (yellow) RNAi-treated populations with accompanying standard error bars (reflecting standard error of the mean) when different time intervals are used for determination of **(A)** total days of health (TDH) and **(B)** healthspan (HS) are shown. Time intervals of >60 seconds are advisable.

of instances, *e.g.* in case of a single bout of activity right before death, after a long period of sickness. In such situations, total days of health (TDH) is a better representation of longitudinal health, but it is also more susceptible to day-to-day variation, *e.g.* classifying a single day of low activity in between several days of obvious health, as unhealthy. Our data show that most HSR values strongly near 1, hence, calculated HS often nearly equals LS (S5 Table). Visual inspection of activity traces from individual worms reveals that in general, HR values better approximate the observed fraction of life spent in a healthy state (S12 Fig). In addition, when comparing the long- and short-lived conditions with controls, HR is capable of distinguishing *daf-2* RNAi-treated populations from internal controls for each experiment, whereas HSR failed to do so in one experiment plate (S6 Table). As expected, neither could distinguish *daf-16* RNAi-treated animals from controls (S6 Table). Based on all these observations, we suggest HR as the primary choice for health readout in experiments where throughput demands fast and simple indicators of potentially interesting conditions.

While TDH and HS are valuable readouts at the level of individual worms, there is complementary value in comparing health at the population level. Additional health information is contained in the shape of the population-level activity curve (Fig 2D), and a metric reflecting this holds added value to the threshold-based TDH and HS. To incorporate this, we also use the area under the population-based survivor activity curve (Fig 11) as a health metric, for ease called integrated activity (IA). For our data, this value is consistently larger for *daf-2* and lower for *daf-16* RNAi-treated populations, as would be expected. Long-lived populations will have larger IA values, despite potentially adding more unhealthy time. Between populations of similar longevity, however, higher IA values reflect interventions that lead to more responsive animals. As opposed to the binary TDH or HS call, IA is assessed along a continuous scale, adding information on the extent of the effect at the population level. In combination with TDH or HS, this more integrative population metric helps distinguish interventions with disproportionate effects on health *vs* longevity.

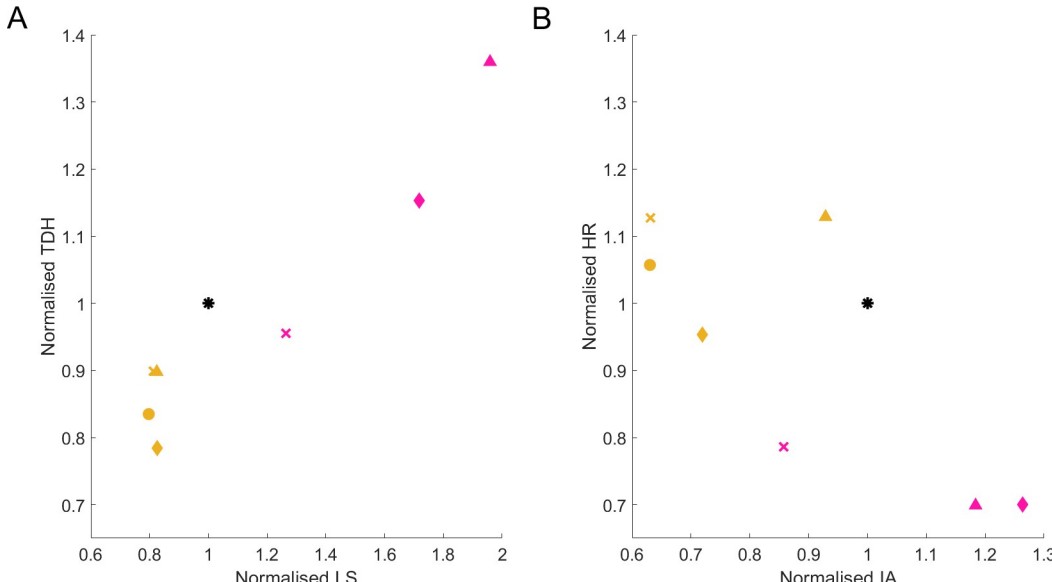

**Fig 11. Normalized IA helps distinguish treated populations from controls. (A)** The mean normalized lifespan (x-axis) *vs* total days of health (y-axis) and **(B)** normalized IA (x-axis) *vs* normalized HR (y-axis), are shown in relation to EV control populations (black, coordinates 1:1:1, all normalizations to internal controls). Average normalized values for populations treated with *daf-2* (pink) or *daf-16* (yellow) RNAi from Experiment I ('◇'), Experiment II ('Δ'), Experiment III ('x') or Experiment IV ('o') are used as coordinates.

The analysis described here aimed to select parameters that can easily distinguish control from long- and short-lived interventions. For this, we relied on the well-described *daf-16* and *daf-2* extremes, but also aim to account for long-lived populations that escape the 'extended twilight' phenotype [13]. It can be concluded that for screening purposes, WorMotel data can be collected at 5s intervals and analyzed at 100s intervals. For fast selection of interesting interventions in settings of considerable throughput, we propose to describe each by a combination of three parameters: lifespan, health ratio (~total days of health) and population-based integrated activity. The WorMotel's ease of use and immense throughput potential balance its larger chip-to-chip variation (in comparison with variation over manual assays), as has been observed for other solutions aiming at increasing throughput [17]. This is why relative assessment, by normalization of individual data to the values of respective control populations, facilitates direct comparison over experiments (Fig 11). As expected, *daf-16* RNAi-treated populations are fairly similar to EV populations but can be distinguished based on integrated activity (Fig 11). *daf-2* RNAi-treated populations show lower normalized health ratios, with ~ 20% decrease of HR mean (Fig 11) and are easily distinguished by a significantly increased LS, decreased HR and increased integrated activity. Conclusions drawn from the here proposed analysis workflow are supported by each individual experiment, hence, are insensitive to inter-experiment variations.

## Discussion

This study aimed to develop a straightforward and user-friendly protocol for rapid identification of interventions affecting aging in *C. elegans*. Our proof of concept study relies on data obtained with the WorMotel [7] and concludes that a variety of interventions can be tested when collecting data at 5s intervals, from which daily peak activity can be calculated using a time interval of 100s. These are the basis for candidate evaluation using three parameters: (1) lifespan, (2) total days of health, and (3) integrated activity, which allow straightforward discrimination of differently aging populations, and keep performing under inter-experimental variation as is typical for these longitudinal experiments [17,25]. Relevant scripts can be found in S2 File.

Collective efforts in the field have revealed pathways capable of determining lifespan of *C. elegans*. However, no perfect consensus can be reached on the concept of being 'healthy', as it is an umbrella term covering many facets of quality of life. Several groups have used different physiological parameters to describe health in *C. elegans*, such as oxidative and heat stress resistance, pharyngeal pumping and autofluorescence [6], vulval integrity [27], intestinal atrophy [28], muscle integrity and yolk production [10]. We here selected daily, stimulated peak activity over daily, stimulated average/median activity as a readout for overall health, as peak values were most robust and less prone to day-to-day activity variations (Fig 5 and S2 Table).

The WorMotel setup uses blue light to stimulate the animals [22,29], therefore peak activity reflects the intrinsic maximal ability of the animal to react to blue light and is evaluated over its lifetime. This organismal response integrates health status of the animal's perceptive abilities with its neuromuscular health. Whereas interventions affecting sensory perception, overall neuronal or muscular health cannot be separated by our approach, previous studies showed no significant difference between survival curves of animals grown in WorMotels (stimulated by blue light) and animals grown in standard plates (stimulated by touch) [7], indicating that the WorMotel provides a good readout of general health in aging populations.

Our analysis revealed that defining the daily activity of a worm by condensing its activity trace of one monitoring period in different ways, delivers slightly different information (S1 and S2 Figs, and S1 Table). The strong correlation of peak with maximal values on one hand,

*vs* of median with integral values on the other, indicates that these parameters together reflect only two interpretations of the stimulated activity trace. Indeed, integral values (~median) take the ability of sustained locomotion upon stimulation into account, whereas peak (~maximal) values are more indicative of the initial ability to respond to the stimulus.

Our definition of health requires individuals to stay above a threshold chosen based on a qualitative analysis of locomotive health in ageing worms. We found that, independent of phenotype, largely non-responding worms—*i.e.* slow or inactive—can be roughly discriminated from healthy animals by applying a pixel difference threshold of 160 (Fig 9, S9 and S10 Figs). While this proved the optimal choice, especially for cases where considerable throughput is expected for data analysis, our data also clearly show overlap between categories, due to large spread within categories (Fig 9 and S11 Fig). This indicates that also for the WorMotel, health remains a noisy concept, and a binary decision (yes or no) is not to be taken as the sole pillar of decision.

In the same line of reasoning, HR deals with limitations regarding the extent to which an animal is healthy. This is why 'integrated activity' is a useful additional metric, even though it can only be used at population level and disproportionally weighs the activities of the longer-*vs* shorter-lived individuals. For populations with similar lifespans and similar HR, IA permits to select the healthier ones. This is especially interesting when looking for interventions that increase health more than they increase longevity, a combination of high biological and medical interest.

Several methods have been reported for automated life- and healthspan evaluation in *C. elegans* [7,16–18,20,21]. While their relevance is evident, widespread use is hampered because these are often the product of in-house optimization. Hence, there are no plug-and-play solutions and the expertise to use such systems is typically contained within only a few individuals globally. With this work, we offer the community an analysis pipeline to easily adopt the WorMotel system as described by Churgin *et al*. [22]. It is our hope that this may facilitate the implementation of automated life- and healthspan evaluation in other labs, as such contributing to progress in the field.

## Supporting information

**S1 Fig. Peak and maximal activity on one hand, *vs* median and integral activity on the other hand, form two separate groups of correlative parameters.** Activity calculated by using the peak values correlates perfectly with maximal activity values (99th percentile) at any time interval **(A)**, whereas correlation with integral **(B)** and median **(C)** values is time interval dependent. **(D)** Integral values, on the contrary, correlate well with median values, but neither of these (**E** integral, **F** median) escape the weaker and interval-dependent correlation with maximal activity. These data suggest that only two interpretations of the activity profile are made by determination of peak/maximal and median/integral daily activities. Time interval dependence of the correlations in B, C, E and F is easily explained by the higher sensitivity of median/integral values to the time interval between analyzed images.
(TIF)

**S2 Fig. Correlation of individual worm data calculated at 100 seconds interval visually shows the two separate groups of correlative parameters.** Activity was calculated based on the different parameters for each worm on each day, independent of genotype. **(A)** Activity calculated by using the peak values correlates perfectly with maximal activity values (99th percentile), whereas correlation with **(B)** integral and **(C)** median values is less pronounced. **(D)** Integral values, on the contrary, correlate well with median values, but both (**E** integral, **F**

median) show a weaker correlation with the maximal value.
(TIF)

**S3 Fig. Peak activity values result in activity curves with the lowest variation, as is clear from the distributions of overall variation based on daily median, integrated, peak or maximal activity for control (black), *daf-2* (pink) or *daf-16* (yellow) RNAi-treated populations across all experiments by increased time interval.** For each individual, day-to-day variation was calculated as stated in the main text. Box values: Q1-2-3, whiskers: +/−2.7σ.
(TIF)

**S4 Fig. Also for individual experiments plates, peak and maximal activity values result in activity curves with the lowest variation.** Distributions of overall variation based on daily median, integrated, peak or maximal activity for control (black), *daf-2* (pink) or *daf-16* (yellow) RNAi-treated populations across all experiments (A Exp I; B Exp II; C Exp III; D Exp IV) follow the same trends as pooled data (Fig 5). For each individual, day-to-day variation was calculated as stated in the main text. Box values: Q1-2-3, whiskers: +/−2.7σ.
(TIF)

**S5 Fig. Variance in *Z*-score is similar for all activity parameters at all time intervals, as is clear from the distributions based on daily median, peak, maximal or integrated activity for pooled control (black), *daf-2* (pink) or *daf-16* (yellow) RNAi-treated populations across all experiments.** For each individual, day-to-day variation was calculated as stated in the main text. Box values: Q1-2-3, whiskers: +/−2.7σ.
(TIF)

**S6 Fig. Variance in *Z*-score based on individual plates is similar for all activity parameters at all time intervals**, as is clear from the distributions based on daily median, peak, maximal or integrated activity for control (black), *daf-2* (pink) or *daf-16* (yellow) RNAi-treated populations across all experiments (A Exp I; B Exp II; C Exp III; D Exp IV). For each individual, day-to-day variation was calculated as stated in the main text. Box values: Q1-2-3, whiskers: +/−2.7σ.
(TIF)

**S7 Fig. Average survivor activity is higher for longer time intervals across all experiments.** Average survivor activity for control, *daf-2* and *daf-16* RNAi-treated populations for (A-C) Exp II, (D-F) Exp III and (G-I) Exp IV. Longer time intervals (≥60s) provide more accurate measurements, this is especially important in late phases of life.
(TIF)

**S8 Fig. The choice of time interval for activity evaluation affects the determination of LS in a genotype-dependent manner across all experiments.** Mean lifespan (error bars: standard error of mean) was calculated for different time intervals for **(A)** Exp I, **(B)** Exp II, **(C)** Exp III and **(D)** Exp IV. The choice of time interval does not affect the calculation of lifespan of control (black) and *daf-16* RNAi-treated (yellow) populations but does affect lifespan decisions made for the long-lived *daf-2* RNAi-treated (pink) populations.
(TIF)

**S9 Fig. A majority of observed pixels changed for healthy worms lie above 177 pixels, whereas the majority of observed pixels changed for unhealthy worm lie under this value.** Determination of a threshold that maximizes the number of truly healthy worms in healthy (very fast—fast—medium fast) categories, while maximizing the number of truly unhealthy worms in the unhealthy categories (slow—inactive) led to a threshold value of 177 pixels

changed. Black line: fraction of animals in the 'slow' category with a pixel difference value < x-axis value; red line: fraction of animals in the 'very fast', 'fast' and 'medium fast' category with a pixel difference value >x-axis value.
(TIF)

**S10 Fig. Threshold determination on individual plate level is very similar to pooled data**, with cumulative curves of from medium fast and slow worms intersecting at approximately 160 pixel differences for (A Exp I; B Exp II; C Exp III; D Exp IV).
(TIF)

**S11 Fig. Decrease in pixel differences with locomotive health follows similar trend for individual plates, with lower pixel differences being assigned to categories of lower locomotive health.** In general, pixel differences below 160 belong to categories 4 and 5 for A Exp I; B Exp II; C Exp III; D Exp IV.
(TIF)

**S12 Fig. Three examples illustrate visually that TDH is a better approximation of health observed in our data when compared to HS.** We plotted the activity profile of three individual wild-type worms whose HS was 1, but whose TDH **(A)** strongly, **(B)** moderately or **(C)** slightly deviated from HS. Activity profile of (A) shows a flare of activity in the last day of life, resulting in a misleadingly high HS. TDH of (B) nears HS more than in case of (A), however, fluctuations in the activity profile of this worm indicate that TDH has a better representation of the animal's health. (C) TDH deviates only two days from the quantified HS, nevertheless, leads to a better approximation of health. All three worms visually indicate that HR (~TDH) is a more accurate quantification of observed health than HSR (~HS).
(TIF)

**S1 Table. Correlation of tested activity parameters depends on the parameter and time interval.** Correlation of the activity values based on either median, peak, maximal (99th percentile) or integral values (see Methods) was tested with a linear regression model for each strain (Column D) and time (column E). $R^2$-values (column C) show that peak activity correlates well with 99[th] percentile value, while median and integral values correlate well with each other. Correlation becomes stronger with increasing time interval.
(XLSX)

**S2 Table. Data based on peak or maximal activity show the lowest overall variation.** Overall variation was calculated for pooled (per genotype) activity data based on either median, peak, maximal (99th p = 99th percentile) or integral values (see methods), as calculated for each time interval (column D). Differences (columns A *vs* B) were compared via Kruskal-Wallis testing (column C: multiple testing-corrected p-values). Bold red: p-values indicative of statistically significant differences.
(XLSX)

**S3 Table. Use of different activity parameters does not influence variance in *Z*-score.** Variance in *Z*-score pooled (per genotype) activity data based on either median, peak, maximal (99th p = 99th percentile) or integral values (see Methods), as calculated for each time interval (column D). Differences (columns A *vs* B) were compared via Kruskal-Wallis testing (column C: multiple testing-corrected p-values). Bold red: p-values indicative of statistically significant differences.
(XLSX)

**S4 Table. Time intervals affect the determination of lifespan in a genotype-dependent manner.** Lifespan was calculated for pooled (per genotype) activity data for each time interval under consideration. Differences (columns A *vs* B) were compared via Kruskal-Wallis testing (column C: multiple testing-corrected p-values). Bold red: p-values indicative of statistically significant differences.
(XLSX)

**S5 Table. HSR values for individual worms often equal 1, independent of genotype or experiment.** LS, HR and HSR values for individual worms for each population and experiment are shown. HSR values often equal 1 for different genotypes, meaning that healthspan and lifespan are the same. HR values on the contrary, reflect more what has been observed in literature with values that often vary from 0.5 to 0.8.
(XLSX)

**S6 Table. Health(span) ratio of *daf-2* RNAi treated animals is significantly different from internal controls.** H(S)R of controls, *daf-2* and *daf-16* RNAi-treated animals were calculated. H(S)R distributions of *daf-2* and *daf-16* RNAi treated animals were probed for significant differences from controls at the same threshold (multiple testing-corrected p-value$_{Kruskal-Wallis}$ < 0,05). Significant p-values are marked in red.
(XLSX)

**S1 File. Pixel difference datafiles for all experiments.** Pixel differences per worm per day were calculated (see Methods) and stored in daily pdata files. Number in pdata file name increases with time. Each column in a pdata file represents an individual worm (from 1 to 240), with each row listing a value according to the time vector.
(ZIP)

**S2 File. Tutorial for data analysis.** This includes relevant scripts (See Tutorial).
(ZIP)

## Acknowledgments

The authors are grateful to Dr. Wouter De Haes for advice regarding statistics, to Ing. Erind Jushaj for assistance with VBA programming, to Bram Cockx for tutorial testing and to Wahab Al-Aani, Elke Vandewyer and Amanda Kieswetter for quality assessment of movement. Strains used in this study were provided by *Caenorhabditis* Genetics Center (CGC).

## Author Contributions

**Conceptualization:** Areta Jushaj, Matthew Churgin, Christopher Fang-Yen, Liesbet Temmerman.

**Data curation:** Areta Jushaj, Matthew Churgin.

**Formal analysis:** Areta Jushaj.

**Funding acquisition:** Areta Jushaj, Christopher Fang-Yen, Liesbet Temmerman.

**Investigation:** Areta Jushaj.

**Methodology:** Areta Jushaj, Matthew Churgin.

**Project administration:** Areta Jushaj, Liesbet Temmerman.

**Resources:** Areta Jushaj, Miguel De La Torre.

**Software:** Areta Jushaj, Matthew Churgin, Bowen Yao.

**Supervision:** Matthew Churgin, Christopher Fang-Yen, Liesbet Temmerman.

**Validation:** Areta Jushaj.

**Visualization:** Areta Jushaj.

**Writing – original draft:** Areta Jushaj.

**Writing – review & editing:** Areta Jushaj, Matthew Churgin, Christopher Fang-Yen, Liesbet Temmerman.

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
