## [Decision Letter · Decision Letter 0]

16 Sep 2019

PONE-D-19-22873

Optimized criteria for locomotion-based healthspan evaluation in C. elegans using the WorMotel system

PLOS ONE

Dear Ms Jushaj,

Thank you for submitting your manuscript to PLOS ONE. After careful consideration, we feel that it has merit but does not fully meet PLOS ONE’s publication criteria as it currently stands. Therefore, we invite you to submit a significantly revised version of the manuscript that addresses all of the points raised during the review process.

The reviewers were split between recommending rejection and major revision.  I believe that the study has merit and should be published, but only after significant revision that addresses each of the reviewers concerns.

We would appreciate receiving your revised manuscript by Oct 31 2019 11:59PM. To enhance the reproducibility of your results, we recommend that if applicable you deposit your laboratory protocols in protocols.io, where a protocol can be assigned its own identifier (DOI) such that it can be cited independently in the future. For instructions see: http://journals.plos.org/plosone/s/submission-guidelines#loc-laboratory-protocols

We look forward to receiving your revised manuscript.

Kind regards,

Sean P. Curran

Academic Editor

PLOS ONE

Journal Requirements:

Additional Editor Comments (if provided):

As you will note, the reviewers identified several major issues with the manuscript. Although the decision to allow a major revision has been made, I want to emphasize that a revised manuscript will need to address all of the reviewers concerns.

Reviewers' comments:

Reviewer's Responses to Questions

**Comments to the Author**

1. Is the manuscript technically sound, and do the data support the conclusions?

Reviewer #1: Partly

Reviewer #2: Partly

2. Has the statistical analysis been performed appropriately and rigorously? 

Reviewer #1: Yes

Reviewer #2: No

3. Have the authors made all data underlying the findings in their manuscript fully available?

Reviewer #1: Yes

Reviewer #2: Yes

4. Is the manuscript presented in an intelligible fashion and written in standard English?

Reviewer #1: No

Reviewer #2: Yes

5. Review Comments to the Author

Reviewer #1: Summary: The paper focuses on optimizing the use of movement-based data from the WorMotel system to better evaluate healthspan. They do this by first evaluating four different statistics of single worm movement before choosing the one they claim is the least variable over independent experiments (peak activity). They then measure “total days of health” and healthspans of daf-2 and daf-16 RNAi animals, both of which are based on varying thresholds of peak activity compared to wildtype. They further analyze these data by using various ratios and integrals to develop further differences between the phenotypes of the animals and to most robustly capture differences in their healthspans.

The principal of these experiments, developing and evaluating statistics to measure worm movement in high-throughput automated systems, is worthwhile. Much of the approach is warranted and well-described. However, some of the tools and many of the quantitative “cutoffs” and measurements are not properly justified. Also, the text does not take confounding issues into account, relying solely upon quantitative movement measurements with little regard to what they might mean and what they might really be measuring. As a whole, the manuscript takes what could be interesting data and makes many assertions that are not well-justified toward the main subject (healthspan/healthy aging) and are not put into a context that makes sense beyond statistical analysis.

Major concerns:

-While pixels changed is a nice quantifiable readout, it is unclear what this represents. It would not be difficult to use the recordings to produce a qualitative corollary (fast/slow movement across plate, partial paralysis, movement in place etc.) and to then quantify these movements as more reliable readouts of movement with pixels changed. The use of an arbitrary quantitative cutoff when qualitative data can be added to it seems like a lost opportunity.

-Calling daily peak activity more robust ignores what these measurements are supposed to take into account. Health is definitely not judged by peak activity (e.g. sprinting), and just because a measurement creates quantitative spread, it doesn’t make it the best measurement. None of these things conclusively measure health.

-Why is it necessary to convert activity to a single value per day? Why not more frequently? Less frequently? Why not multiple values? It seems like they are just trying to make it simple for simplicity and not for a well-justified reason.

-All of the choices presented for single value readouts seem to have more random variation than one would like, reflecting higher day-to-day oscillations and thus challenges in the readout. This is a function of the complexity of complex organisms.

-Is peak response really the maximum ability to move? Or does it also have to do with sensitivity to the blue light changing with age and a number of other unknown factors? None of the other potential factors are mentioned or accounted for.

-“Smoother curves lead to more robust assessment.” They may lead to more separation, but that doesn’t make it necessarily correct. It is looking at different things, some of which have different variation, and none of which are looked at for why they vary and thus why they might be better or worse indicators of health. It is impossible to say what is better or worse at measuring health, only that some are statistically more separable.

-What is a “non-optimal activity?” How can one assess this? What does this mean? Without qualitative measurements this is a completely arbitrary cutoff with no real meaning.

-What exactly does “total days of health” tell you? They define the parameter, but how does it contribute to a better description of the health of the animal? It seems they use it primarily because it is less than the healthspan (in numerical quantity) resulting in better resolution of the health ratios they utilize later in the paper. If the healthy days don’t have to be consecutive, what does it mean when a worm doesn’t have a “healthy” day, but resumes being healthy the following day? HR “outperforms” HSR as a distinguishing health metric? What does this mean? Arbitrarily creating increased differences is not outperforming. It is not justified why TDH is a useful metric, what it really means, or whether it has any place in measurements.

-“The WorMotel setup uses blue light to stimulate the animals therefore peak activity reflects the maximal ability of an animal to move, and it is evaluated over its lifetime” This may be true in young worms, but is not known in older individuals who lose not only the ability to move, but to perceive things such as blue light.

Minor concerns:

Figure 3: Peak activity and 99th percentile appear do have the lowest variation but no statistics that they are significantly different than median or integrated activity are given.

Figure 6B, D: B has no error bars (or maybe they’re just really small) while D has huge error bars. In general, dividing both TDH and healthspan by the same value (lifespan) is plotting the same information on a different scale. This seems unnecessarily redundant since the trends are going to be the same.

Line 315: They propose 100 seconds as the “ideal compromise” for the time intervals, but if they identified 60 – 300 seconds a finer gradient would be necessary to find a more optimal time instead of just taking the middle time point between the two.

Figure 7: They never actually define the acronym IA in the paper. A) The variation in their individual daf-2 RNAi experiments makes it hard to determine the significance of the points compared to WT. B) It would seem that the daf-16 RNAi animals are healthier but move slightly less than WT over their lifespans, while the daf-2 RNAi animals are less healthy but move more than WT over their lifespans.

The daf-2 RNAi animals are healthier (i.e. they move more) at later stages in life (the “twilight” period) than WT, but since their activity is below the threshold chosen by the authors, daf-2 RNAi animals are considered to be less healthy than WT. This illustrates the problem with arbitrary thresholds: depending where they are set you can get either answer.

Figure S6: It appears the plots are reversed compared to what they describe in the caption.

Discussion:

Line 355: “…a wide variety of interventions can be tested…” Only one type of was tested here. This seems like an overreach.

Reviewer #2: SUMMARY

The authors use data from the previously described "WorMotel" system to evaluate metrics for assaying health and healthspan. In particular, the authors compare N2, daf-2(RNAi) and daf-16(RNAi) for health effects under a variety of metrics to identify an "optimal" scheme for measuring individual and population health (according to several metrics the authors have chosen). The authors find that peak activity across measurement intervals of 60-300 sec. provide the least variable measurements of movement among those considered. They further argue that two metrics provide useful and complementary information about health: indivdual health ratios (a ratio of a healthspan-like metric to lifespan) and population-level integrated activity.

IMPRESSION

There is great experimental and conceptual inconsistency within the field regarding how to evaluate "healthspan" in C. elegans. The current study provides a useful starting point for a much-needed discussion about experimental design and data analysis.

Before that, however, there are several points within the study where the authors need to clarify methodological details, and explicate some of the logic by which they draw their conclusions. In particular, the criteria by which the authors evaluate "optimal" metrics are rather ad hoc and deployed with little justification or statistical backing. More explanation here will be crirical.

In related matters, the authors reasoning in justifying their choice of health measures is somewhat circular in places. The main logic appears to be that "a good healthspan metric should distinguish daf-2, daf-16, and control conditions" -- which presumes that healthspan is actually different in those cases. What if the "true" healthspan (or normalized healthspan ratio) is actually the same in daf-16 vs. N2? There's no a priori reason to assume that the "right" metric is one that separates those conditions. At a minimum these caveats need to be carefully discussed.

The writing is also somewhat loose and informal throughout, and in places overly vague regarding specific experimental details. Nevertheless, with a little tightening of both the writing and the logic of the data analysis -- and perhaps dialing some of the conclusions back to what is supported by the data -- this will be a useful contribution to the literature.

MAJOR CONCERNS

1) In the abstract, the authors refer to RNAi against daf-2 and daf-16 (vs. empty-vector control) as a "wide range of conditions". Indeed, throughout the work the authors simply assume that any findings based on analysis of these three conditions will inherently generalize to other (non-IIS-perturbing) conditions. This is a bit of a stretch, and the authors need to be much more careful with claims of generality after examining only IIS-pathway perturbations.

2) Many of the methods are under-described, especially for a manuscript that aims to propose a canonical analysis scheme.

2a) The authors need to describe precisely how image pixels are turned into numerical scores. I assume the image immediately before stimulation is compared to images at different intervals after stimulation. (Or are the differences calculated not for t=0 vs. t=n, for all n, but for t=n vs. t=n+1? This matters, obviously.) Next, I assume a pixel-wise difference image is calculated for whichever pair of images is under consideration. But how is that image then converted into a single difference score? The median absolute pixel difference? The root mean squared pixel difference? The sum of absolute differences? The count of pixels that are different by a certain threshold? (If the latter, how is the threshold chosen? And how does the threshold choice influence subsequent conclusions regarding healthspan?)

If a sum-of-differences type of metric is used (rather than e.g. a count-of-above-threshold-differences), the authors discuss the caveat that conditions that change that pixel intensity distribution of worms (i.e. produce individuals that are either more clear or darker or more mottled than WT) will naturally generate different pixel differences for the same amount of total movement. This is also a problem for thresholded counts, but may be less severe depending on the threshold employed. Overall, these details matter, and they need to be explicitly described in the methods (rather than be left implicit in a matlab script) and the choices / trade-offs should ideally be justified if possible.

2b) Given a pixel-difference score as a function of time post-stimulation, the authors next propose several ways to summarize that as a single number for each individual. Of these, the "peak activity" score needs to be described and justified better. What does "average of the 95th to 99th percentiles" mean? Is it the mean of all values within the range defined by the 95th to 99th percentiles? Or just (95th-percentile-value + 99th-percentile-value)/2? More generally, what is the point of using both a percentile and an arithmetic mean? The choice is justified as saying it "should be slightly less prone to outliers or noise", but I'm not sure on what statistical basis one might conclude that a mean (which is outlier prone) of a percentile (which is more robust to outliers) would be better than just e.g. using the 97.5th percentile or whatever. Especially given that this mathematically-weird metric is the one the authors recommend later, a little more description / justification is warranted.

2c) The overall variation score is not clearly defined. Presumably the brackets in the denominator of the numerator of the overall formula represent the expectation (mean) of the day-to-day absolute changes. But what is the mean over -- is it a mean of all individuals at that day? A mean of all days for that individual?

Next, in the results, this is described as a sum of changes "in percent", but it's not clear that the formula really is calculating percentage changes in any meaningful way. A percent (or fractional) change would generally be calculated something like |a_i - a_i+1| / |a_i|, or similar. (I.e. change / baseline, rather than the current formula, which is individual-change / population-mean-change.)

More generally, this score is rather ad hoc, and the specific choice isn't particularly well justified. In particular, given that some degree of activity-score changes over time is expected (as aging happens, scores decline), clearly an optimal health score isn't just one that minimizes all changes over time!

The authors are correct in their desire for a metric that changes smoothly, rather than one that is noise-prone and gives jagged curves over time. But the authors' proposed variation score doesn't just penalize jagged curves -- it penalizes any curve that isn't completely flat. That is, the only way to minimize the proposed total variation score is for a health measure to give a completely flat, zero-slope line. So, broadly, the key metric for this whole manuscript isn't really measuring what the authors want it to measure (or claim it is measuring). Intuition: jaggedness is a second-derivative (and higher derivatives) property, but the score is just examining first derivatives.

A better approach might be to minimize day-to-day changes with reference to the overall population trend. Something more like |a_i/mean(a_i) - a_i+1/mean(a_i+1)| (where the mean is across all individuals at that timepoint), for example. A good health measure could produce zero on this variation score and still allow for smooth changes in health over time.

Even that's pretty ad hoc, however, and would confounded by changes in the population variance in activity scores over time. So the real right answer is for the authors to look to the existing statistical toolkit, rather than reinventing their own measures of variation. In particular, if the authors wish to measure variation, then statistical variance would be an obvious choice. But one wouldn't want to measure the variance of the a_i since, as above, those scores are expected to change over time. A more principled approach would be to calculate each individual's movement z-score as a function of time i.e. z_i = (a_i - mean(a_i)) / std(a_i) (where the mean and std are across all individuals at that timepoint), and then calculate the variance of the z_i scores across all timepoints for that particular individual. The "smoothest" activity measure would be the one with the least variance in z-scores over time.

Note also the "peak activity" metric is truly the best and most robust, one might hope that it would perform best across a panel of different metrics, such as all those suggested above. But if each different variation score yields a different choice of "best metric", one might become skeptical of the whole project.

2d) A "5 pixels changed" threshold is used to define lifespan (line 146-7 and 265-6), but there is no discussion of (i) how this threshold was chosen, (ii) over what time interval it is computed (since the authors note that longer time intervals increase sensitivity to movement), (iii) how sensitive lifespan estimates are to that specific threshold.

3) Regarding the different activity metrics (peak, median, etc.), how much do these distinctions actually matter? Do any of the conclusions from the later sections actually depend on using peak/99th percentile, or are the qualitative results the same if median or integral was employed? For that matter, how well correlated are all of these measures with one another? Perhaps this is all a bit of a distinction without a difference, if all the measures are highly correlated...

4) Regarding the determination of an appropriate healthspan descriptor (HR vs. HSR) + threshold (lines 273-299), the logic/language in this section is pretty unclear throughout.

4a) Consider making these paragraphs into a new section altogether (e.g. “determining ideal metrics for health”).

4b) It would be useful to clearly state ahead of time that the decision of interest involves choosing HR vs. HSR (as opposed to TDH vs. HS, and separate from just determining the threshold).

4c) lines 273-289 and lines 290-299: the authors appear to be making the assumption that the ideal descriptor and threshold should (i) distinguish among daf-2, daf-16, and WT, and (ii) be consistent across experiments. The authors should state these criteria explicitly and justify them. In particular, as above, this introduces some degree of circular reasoning. If the authors chose their health metric based on how well it can distinguish daf-2 and daf-16 from WT, then some reader of this paper would be on very shaky ground to then use that metric to try to learn anything about the health effects of IIS by using such a metric.

4d) lines 292-295: "at very high thresholds, nTDH reflects the intrinsic activity of the young worm populations (vs control levels), whereas at low thresholds, this value approaches longevity (Fig S4 and S5). The optimal threshold region can be found in between these two, where HR (=TDH/LS) reflects health". The authors' argument seems to be that because the extremes of threshold ranges are bad health scores, the middle of the range must be a good score. This is not a priori true: the middle of the range could be bad too! Absent some external criterion, there's no way for the authors to really figure out what "optimal" might be.

5) The authors don't mention anywhere the real take-home message of Figure 6, which is that the HSR metric is hugely more variable than all the rest (look at those standard errors!) and is probably useless as a result.

6) The discussion of the integrated activity (Figure 7 and lines 320-327) is extremely vague and unclear: what does the integral of the population curve tell us that the population averages of the individual statistics don't? The figure legend claims that the normalized IA better distinguishes the daf-16 replicates from WT (and/or daf-2) than the other measures, but that's not visually obvious from Figure 7, nor is there any quantification.

7) Lines 343-345: "This is why relative assessment, by normalization of individual data to the values of respective control populations, facilitates direct comparison over experiments (Fig 7)." This is an assertion, but isn't really backed up by the data. Figure 7 still shows a ton of replicate-to-replicate variability. How much smaller is the normalized inter-replicate variability compared to un-normalized? (At high enough levels of inter-plate variability, normalization can't really help anymore because you can't even count on the experiment-control pairs to be comparable within a replicate.)

8) Lines 377-379: "In line with literature, this classifies the ‘gerospan’ or ‘extended twiglight’ of daf-2-like interventions as unhealthy". This isn't really consistent with the authors' data, which show that daf-2 has a larger TDH, HS and IA than WT. The HR is smaller for daf-2, and the HSR is so variable as to be useless (as above). Thus across many of the authors own measures daf-2 is perfectly healthy. The "deficiency" of daf-2 is that while it gains many more days of good health vs. WT, it also gains even more days of poor health. Whether or not this is a "good" tradeoff is a value judgement rather than an empirical fact. Moreover, as per Hahm 2015, under other metrics of health daf-2 behaves quite differently.

Related to the above, lines 384-385 are also an assertion / value judgement masquerading as fact: "interventions that increase health more than they increase longevity [are] a combination of high biological and medical interest". Take for example a condition in which daf-2 individuals are euthanized the second that their activity falls below whatever threshold is defined for "good health". This condition would produce a substantial increase in health with a much more modest increase in lifespan (i.e. this would truncate the "extended twilight" tail of daf-2). Would such an intervention really be better or more medically interesting than a daf-2-like intervention that doesn't involve euthanasia? The fact that lifespan-limiting "interventions" can trivially produce health ratios of 1 suggests that the HR may not really be the most biologically informative measure of "good health". Some careful discussion of what health ratios actually mean (if anything) is in order.

MINOR CONCERNS

1) Typos or vague / unclear language :

lines 71-75: run-on / complex / overly informal sentence.

line 139: “day-tot day”

lines 218-221: what is actually being said here?

lines 278-281: likewise unclear.

line 378: “twiglight”

2) More care should be taken in referencing:

2a) ref 12, the Lifespan Machine paper, is not particularly germane to the claim on lines 171-173 that "The exact effects of these genetic interventions on the lifespan of C. elegans varies somewhat in high-throughput screens and between different labs".

2b) The following aren't in the correct format / style:

line 50: Bansal et al.

line 89: Wormbook

2c) Probably should also cite Herndon 2002, Huang 2004, Pincus 2011, and Zhang 2016 regarding movement decline as a predictor of lifespan (line 52). Likewise, citations for longitudinal analysis of individuals / populations are a bit spotty. Hulme 2010, Pincus 2011, Zhang 2016 would be good to include for the individual case, and both of the Stroustrup papers for populations.

2d) line 113-115: Citing Hahm 2015 here is deceptive. They showed that spontaneous movement on food is confounded by genotype, compared to maximum velocity off food. But the Hahm results have nothing at all to say about whether stimulated movement (on food) is more or less confounded than spontaneous movement (on food), which is the point that the present authors are attempting to make. If anything, the original Churgin paper may better support this statement. If this is an important of a point to emphasize, the authors should explain their rationale more carefully.

3) Lines 232-234: "Based on these considerations, we opted for the peak activity as the activity value of choice, preferring it over the maximal activity solely based on the fact that the first value is based on more observation points and might therefore be a more accurate representation of true biology." This is not really a mathematically or statistically cogent statement. Calculating a percentile uses all of the input data in the same way that calculating a mean does. It doesn't somehow use "more" data to calculate a percentile and then calculate a mean based off of a subset of the data defined by the percentile.

4) Figure 2: (a) Consider labeling titles/axes for each respective graph to identify which statistic is used to generate the data. (b) Panels B & C for 99th and peak activity appear identical; check whether these graphs have been correctly generated.

5) Figure 5: Why do daf-2 lifespan estimates decrease with the longest time intervals? Are the animals somehow returning back to where they started such that the difference scores decrease and they register as "dead" incorrectly? This is a more than a little odd...

6) Figure 7: It would be helpful to explicitly mention that (a) everything in this figure is normalized to the mean of WT, and (b) IA stands for integrated activity. (Including in the relevant methods section.)

7) Though mentioned in Methods, consider reintroducing the concept of HR/HSR in the Results in greater detail than lines 255-6. These terms are heavily used in the Results and it is a bit be difficult for the reader absent a refresher.

8) Table S4: Consider making the second column title for Table S4 “p-value for interaction with experiment”

6. PLOS authors have the option to publish the peer review history of their article (what does this mean?). If published, this will include your full peer review and any attached files.

Reviewer #1: No

Reviewer #2: No

---

## [Author Response · Author response to Decision Letter 0]

24 Dec 2019

See submitted file ' Response to reviewers'.

---

## [Decision Letter · Decision Letter 1]

11 Feb 2020

Optimized criteria for locomotion-based healthspan evaluation in C. elegans using the WorMotel system

PONE-D-19-22873R1

Dear Dr. Jushaj,

We are pleased to inform you that your manuscript has been judged scientifically suitable for publication and will be formally accepted for publication once it complies with all outstanding technical requirements.

With kind regards,

Sean P. Curran

Academic Editor

PLOS ONE

Additional Editor Comments (optional):

Reviewers' comments:

Reviewer's Responses to Questions

**Comments to the Author**

1. If the authors have adequately addressed your comments raised in a previous round of review and you feel that this manuscript is now acceptable for publication, you may indicate that here to bypass the “Comments to the Author” section, enter your conflict of interest statement in the “Confidential to Editor” section, and submit your "Accept" recommendation.

Reviewer #1: All comments have been addressed

2. Is the manuscript technically sound, and do the data support the conclusions?

Reviewer #1: Yes

3. Has the statistical analysis been performed appropriately and rigorously? 

Reviewer #1: Yes

4. Have the authors made all data underlying the findings in their manuscript fully available?

Reviewer #1: Yes

5. Is the manuscript presented in an intelligible fashion and written in standard English?

Reviewer #1: Yes

6. Review Comments to the Author

Reviewer #1: This revised manuscript focuses on optimizing the use of movement-based data from the WorMotel

system to better evaluate healthspan. They do this by first evaluating four different statistics of

single worm movement before choosing the one that is the least variable over independent

experiments (peak activity). They then measure “total days of health” and healthspans of daf-2 and

daf-16 RNAi animals, both of which are based on varying thresholds of peak activity compared to

wildtype. They further analyze these data by using various ratios and integrals to develop further

differences between the phenotypes of the animals and to most robustly capture differences in their

healthspans.

I would like to thank the authors for their revised manuscript. In the revised version, the authors have done a tremendous job in responding to the reviewer comments, and as such, the paper is much improved. Based on the improved explanations and additional data, I now see this manuscript as an extremely useful text for scientists interested in C elegans aging and healthspan to refer to and to move the healthspan conversation forward. My only note to the authors, based on this revised manuscript, is that some figures (e.g. Fig 6, 7, 9) lack clear labeling where it would be helpful in the figure, and some images are quite blurry. This likely can be fixed during production.

7. PLOS authors have the option to publish the peer review history of their article (what does this mean?). If published, this will include your full peer review and any attached files.

Reviewer #1: No

---

## [Editor Report · Acceptance letter]

24 Feb 2020

PONE-D-19-22873R1 

Optimized criteria for locomotion-based healthspan evaluation in *C. elegans* using the WorMotel system 

Dear Dr. Jushaj:

I am pleased to inform you that your manuscript has been deemed suitable for publication in PLOS ONE. Congratulations! Your manuscript is now with our production department. 

With kind regards,

on behalf of

Dr. Sean P. Curran 

Academic Editor

PLOS ONE